# Factors Influencing the In Vitro Maturation (IVM) of Human Oocyte

**DOI:** 10.3390/biomedicines9121904

**Published:** 2021-12-14

**Authors:** Huixia Yang, Thomas Kolben, Sarah Meister, Corinna Paul, Julia van Dorp, Sibel Eren, Christina Kuhn, Martina Rahmeh, Sven Mahner, Udo Jeschke, Viktoria von Schönfeldt

**Affiliations:** 1Department of Obstetrics and Gynecology, University Hospital, Ludwig-Maximilians-University, 81377 Munich, Germany; huixia.yang@med.uni-muenchen.de (H.Y.); thomas.kolben@med.uni-muenchen.de (T.K.); sarah.meister@med.uni-muenchen.de (S.M.); corinna.paul@med.uni-muenchen.de (C.P.); julia.van.dorp@hotmail.de (J.v.D.); sibel.eren@med.uni-muenchen.de (S.E.); christina.kuhn@uk-augsburg.de (C.K.); martina.rahmeh@med.uni-muenchen.de (M.R.); sven.mahner@med.uni-muenchen.de (S.M.); viktoria.schoenfeldt@med.uni-muenchen.de (V.v.S.); 2Department of Obstetrics and Gynecology, University Hospital Augsburg, 86156 Augsburg, Germany

**Keywords:** in vitro maturation (IVM), biphasic IVM, antioxidant, age, vitrification

## Abstract

In vitro maturation (IVM) of oocytes is a promising assisted reproductive technology (ART) deemed as a simple and safe procedure. It is mainly used in patients with impaired oocyte maturation and in fertility preservation for women facing the risk of losing fertility. However, to date, it is still not widely used in clinical practice because of its underperformance. The influencing factors, such as biphasic IVM system, culture medium, and the supplementation, have a marked effect on the outcomes of oocyte IVM. However, the role of different culture media, supplements, and follicular priming regimens in oocyte IVM have yet to be fully clarified and deserve further investigation.

## 1. Introduction

In vitro maturation (IVM) of oocytes is a technology with a long history. As early as the 1930s, in vitro matured oocytes were utilized in the initial attempts of human in vitro fertilization (IVF) [1,2,3,4]. In 1965, Edwards demonstrated that human oocytes collected from antral follicles could be matured and fertilized in vitro [5,6]. In 1991, Cha and colleagues reported the live birth after IVM of oocytes from non-stimulated cycles followed by IVF, leading to the clinical investigation of IVM [7]. In 2013, it has been estimated that more than 5000 babies derived from IVM have been born around the world [8].

IVM is primarily used in patients with polycystic ovary syndrome (PCOS) or poly-cystic ovaries (PCO) and patients at risk of ovarian hyperstimulation syndrome (OHSS) [9]. For the patients, this technology is cost-effective, safe, and convenient (fewer hormone injections, shorter stimulation cycles, avoidance of OHSS, reduced frequency of monitoring); for the IVM offspring, the initial studies of small sample size showed no significant differences in obstetric, perinatal, and developmental outcomes between IVM and conventional IVF [10,11,12,13,14,15], though large-sample-size long-term follow-up is necessary to comprehensively assess the offspring developmental outcomes. As IVM’s most remarkable advantage is its minimal or even zero ovarian stimulation, it is a good choice for the conditions in which hormone stimulation is contra-indicated or hyper-responsive; it is also a good alternative choice for women who want to donate/preserve oocytes but do not want to suffer side effects from hormone stimulation. With the development of IVM in recent years, the clinical indications for IVM treatment have expanded to (1) fertility preservation (FP) for women at risk of fertility loss and women seeking planned oocyte cryopreservation, (2) women with high/poor response to exogenous gonadotropin treatment, (3) oocyte donation, (4) patients with thrombophilia, and (5) previous failed IVF attempts/recurrent implantation failure [16,17,18,19,20].

However, to date, IVM is not widely applied in clinical practice because of its inferior performance (e.g., low rate of oocyte retrieval and maturation, low rate of blastocyst conversion, and low rate of implantation and pregnancy) compared to conventional IVF [21]. Besides, IVM outcomes among different centers are variable due to a lack of standard and effective protocol for IVM [22]. The definition of IVM also varies among different scholars [5,23,24,25,26,27,28]. Based on the most recent document from the American Society for Reproductive Medicine (ASRM), the clinical human IVM refers to in vitro maturation of the immature oocytes retrieved from follicles after no follicle stimulating hormone (FSH) or minimal FSH stimulation (usually 3 days’ stimulation), followed by no human chorionic gonadotrophin (hCG) or minimal hCG priming (single 10,000 IU injection) [21].

Knowledge about the influencing factors of oocyte IVM is essential for improving its outcome. In the current review, factors including (1) follicular priming methods and collected eggs, (2) culture conditions, (3) culture methods, (4) time of oocyte retrieval, (5) time of in vitro culture, (6) woman’s age, and (7) cryopreservation and other physical factors were discussed to enable a comprehensive discussion. All these factors may have an impact on the developmental potential of oocytes and/or the synchronization of nuclear and cytoplasmic maturation.

## 2. Follicular Priming Methods and Collected Eggs

Table 1 is the summary of the studies on the impact of follicular priming methods on human oocyte IVM outcomes. IVM is usually performed in an unstimulated cycle, or a cycle with minimal gonadotropin stimulation to collect immature cumulus-oocyte complexes (COCs). Based on the published reports on clinical IVM, as well as the recent document from ASRM, the follicular priming methods include low doses of FSH stimulation, single hCG injection, and minimal FSH stimulation before single hCG injection [21,26,27,29,30,31,32]. Based on the study of Fadini et al. [27], after single hCG priming or FSH combined with hCG priming, it is possible to collect the already in-vivo-matured oocytes.

FSH is a frequently used gonadotropin for inducing follicle development and ovulation in ART. Generally, in a single FSH-primed IVM cycle, low doses of FSH are used to minimally stimulate ovaries for 3–6 days, followed by oocyte retrieval on days 7–10 without hCG administration. Mild FSH-primed IVM cycles for non-PCOS women with proven fertility [26] and PCOS women with at least one year of infertility [32] have been proven to increase oocyte maturation rate compared to non-stimulated IVM cycles. However, another study revealed that, compared with the unstimulated group, single FSH stimulation in the naturally cycling non-PCOS women did not improve the maturation and fertilization rate of oocytes [29]. Several factors could have contributed to the conflicting results between [26,29,32]. Comparing the study by [29,32] reveals they were involved in different study populations, and it is possible that single FSH priming may be more suitable for PCOS patients rather than naturally cycling non-PCOS women in regard to IVM, which needs to be validated through further investigation. For the studies by [26] and by [29], even though they focused on similar study populations, the oocytes in [26] were primed with a truncated course of FSH (day two: 300 IU FSH; day four: 150 IU FSH; day six: 150 IU FSH), and the culture medium in [26] was serum-free and not a commercial IVM medium. Therefore, future studies are warranted to clarify the influence of FSH stimulation on IVM outcomes.

The practice of hCG priming in IVM is a highly debated topic and seems to be double-edged. On the one hand, it has been revealed that the use of hCG can enhance angiogenesis of the endometrium [33], promote endometrial receptivity [34] and blastocyst implantation [33], and thus is needed in fresh IVM cycles [34]. A large-scale randomized controlled trial (RCT) involving 400 normo-ovulatory women demonstrated that FSH + hCG priming could significantly increase the IVM rates of oocytes compared to single FSH-priming, hCG-priming, or non-priming [27]. On the other hand, in recent years, scholars have proposed that the definition of IVM [23,24] can include the cycles primed with FSH or its analogues, but should exclude cycles that are intended to trigger oocyte maturation in vivo, such as hCG. Indeed, hCG priming might accelerate the maturation of oocytes in vivo, and make the obtained oocyte population heterogeneous (i.e., being at different stages of maturation). This situation requires a higher frequency of oocyte monitoring. Despite that, some pre-mature oocytes may still undergo an excess in vitro culture time, which makes these oocytes begin aging and pass the optimal fertilization time. Collectively, in the authors’ opinion, if the women undergoing IVM do not require embryo transfer for themselves in a fresh cycle, then hCG priming should be avoided.

## 3. Culture Conditions

### 3.1. Culture Medium

The culture medium used in oocyte IVM is the bedrock of the IVM cycle. Different constituents in the IVM medium can influence IVM of the oocyte and subsequent embryo development [35,36]. To date, the reported culture media for IVM in research or clinical applications include several common media and two commercial tailor-made media for IVM. However, no perfect culture medium for IVM has been established yet.

Common media mainly include the tissue culture medium 199 (TCM-199), human tubal fluid (HTF), Ham’s-F10 medium, glucose-free medium (P1), Chang’s medium, and blastocyst medium [37,38,39]. Based on existing literature, TCM-199 is the most commonly used non-tailor-made medium for human oocyte IVM [40,41,42,43,44,45,46,47]. It has also been reported that the performance of TCM-199 in human oocyte IVM is superior to HTF with respect to maturation rate, fertilization rate, and embryo quality [39]. However, another study revealed that the glucose-free P1 medium is superior to TCM-199 (which contains a high glucose concentration) for IVM of human granulosa cell-free oocytes [9]. Commercial tailor-made IVM media mainly include MediCult IVM medium and SAGE IVM medium. These two media have been shown to be equally efficacious [48]. Most previous studies have revealed that the laboratory outcomes in IVM were similar between commercial IVM media (MediCult/SAGE) and other common medium (e.g., TCM-199 medium [49], blastocyst medium [50,51], cleavage medium [51], and homemade IVM medium [51]). There is also a study which revealed that the blastocyst medium enabled a higher IVM rate than the commercial IVM medium [52].

Table 2 is a summary of the influence of culture medium on human oocyte IVM. In general, the published studies only compared two or four media types in their investigations, and most of the comparisons were statistically non-significant regarding the IVM outcomes of oocytes. Admittedly, the egg sources, priming methods, medium supplements, and in vitro culture time varied among studies, so it is not meaningful to perform a direct comparison of the efficiency of various culture media across different studies. Further studies are needed to perform a more comprehensive comparison using diverse culture media. It is also warranted to develop better commercial IVM culture media with minimal batch-to-batch variation that enhances oocyte IVM rates and subsequent embryo development.

### 3.2. Protein Sources in Culture Medium

Protein sources are the most common supplement to IVM medium, providing a nitrogen source and acting as an antioxidant, as well as chelator of toxic metal ions [54]. Most media require the addition of a protein source. The exceptions are the commercial SAGE-IVM medium and the LAG medium from MediCult IVM System, which already contain synthetic serum replacements, therefore, no extra protein source is required [48].

During the early development of IVM technology, fetal cord serum (FCS) [7,55], or human follicular fluid (HFF) [7,55], or human peritoneal fluid (HPF) [55] was added to human oocyte IVM medium. Later, patient’s own serum [48], or human serum albumin (HSA) [40,41,42,56,57], or synthetic substitutive serum (SSS) [9,39], or other synthetic serum substitutes [43] were used as a protein source supplement in the maturation medium. Considering the complex mixtures, batch-to-batch variability, and the potential risks of disease/infection transmission with protein supplements from biological sources, HSA and other synthetic serum substitutes are recommended for protein supplementations, as their compositions are relatively pure and well-defined.

Previous study revealed that IVM media supplemented with HSA or HFF resulted in similar developmental competency in immature oocytes [54]. However, another study found that, compared with HSA-supplemented media, patients’ own-serum-supplemented media produced significantly higher maturation rates in oocytes (41% vs. 63%) [58]. This finding indicates that the substances, excluding albumin, presented in maternal serum can influence the maturation of human oocytes. However, studies are needed to develop the ideal protein source that adequately supports human immature oocytes and replaces the maternal serum.

### 3.3. Hormones in Culture Medium

The oocyte IVM systems are hormone dependent. Hormones, such as recombinant FSH, hCG, luteinizing hormone (LH), insulin, and estradiol (E2), have been added to the IVM medium. However, choosing hormonal supplements and their corresponding concentrations are variable and controversial. FSH is the most basal hormonal supplement in the IVM medium. It can be used as the single hormonal supplement in IVM medium [59,60]. It has been revealed that the addition of FSH supports nuclear maturation but postpones meiotic progression, playing a positive role in oocyte cytoplasmic maturation [61]. In existing literature, the concentration of FSH added to IVM medium ranged from 0.075–0.75 IU/mL, where 0.075 IU/mL was the most widely used concentration. The highest concentration at 0.75 IU/mL did not seem to play any role in enhancing IVM rates [20,29,62,63,64]. In most studies, FSH was usually used in combination with LH [43,56,65,66]/hCG [40,41,44,45,46,67].

Nevertheless, the role of LH and hCG in the IVM medium is controversial. It was previously observed that recombinant hCG (r-hCG) and recombinant LH (r-LH) were equipotent in promoting the IVM of human oocyte [68]. On the contrary, a mouse study reported that the addition of purified urinary hCG (u-hCG), but not r-LH, to IVM medium significantly increased maturation rates of mouse immature oocytes. Compared with u-hCG, r-LH was found more favorable for early embryonic development [69]. Animal mechanism studies indicated that the addition of LH to IVM medium may not be necessary for oocyte IVM [70,71,72], since the levels of LH receptors in cumulus cells (CCs) is very low [73]. In mammalian ovarian follicles, LH promotes and regulates oocyte maturation by acting on the outer granulosa cells, and not the COCs [74].

### 3.4. Carbohydrates in Culture Medium

Carbohydrates, such as glucose, lactate, and pyruvate, are substrates for generating ATP during oogenesis and have a stimulatory action on the nuclear maturation of oocytes [75]. The oocyte itself has a relatively poor capacity to metabolize glucose, rather, glucose in medium is consumed by CCs, and evidence has shown that CCs are responsible for providing the intermediates of glucose metabolism, such as pyruvate and lactate, which are the preferred energy substrates of the oocyte [76]. While animal studies revealed that, compared with glucose and pyruvate, follicular cells did not particularly rely on lactate for their energy requirements [77], lactate may, however, have additional, non-metabolic roles in the acidification of the local environment [78].

For cumulus cell-enclosed oocytes, media with glucose alone added as an energy source could enable the oocytes to resume meiosis, while for the cumulus cell denuded oocytes, glucose alone does not work, and supplementation of pyruvate in culture media is essential for the resumption of meiosis in denuded oocytes [79]. In a study by Cekleniak et al. [9], though lacking a significant *p* value, a glucose-free P1 medium (sodium pyruvate: 36.30 mg/L; sodium lactate: 2399.00 mg/L; and glucose: 0 mg/L) brought out higher maturation rates in human cumulus-striped oocytes than a high-glucose TCM-199 medium (sodium pyruvate: 0 mg/L; sodium lactate: 0 mg/L; and glucose: 1000.00 mg/L). This study suggests that, in the absence of CCs, a higher oocyte maturation rate may be achieved in a culture medium with pyruvate and lactate added but without or with little glucose.

### 3.5. Other Supplementation in Culture Medium

Except for the proteins, hormones and carbohydrates, other supplementations such as cytokines, antioxidants, and CCs have also been investigated regarding their influence on the efficiency of IVM as summarized in Table 3. The addition of antioxidants to human oocyte IVM medium has been investigated by multiple studies [41,44,45,46,80]. Most of the studies demonstrated a positive role of antioxidants (e.g., Coenzyme Q10 [80], melatonin [41], quercetin [45], resveratrol [46]) in improving the IVM rate of human oocytes. Compared with the in vivo environment, the in vitro culturing of oocytes lacks a natural antioxidant system, which may cause excess reactive oxygen species (ROS) generation, resulting in oxidative stress [81]. The antioxidants in the culture medium may serve as ROS scavengers. Nevertheless, the improvement of culture condition is a complex challenge that depends not only on the choice of an antioxidant but also on its concentration. The final effect of antioxidant addition should be keeping the ROS within physiological range and maintaining redox balance.

In addition, to maintain cellular morphology and function similar to in vivo, the concept of a three-dimensional (3D) culture system has been proposed. This system cultures the cells in collagen, matrigel, fibrin [82], or other biomaterials like 3D scaffolds which can provide a 3D microenvironment for cell growth. Torre et al. developed a follicle-mimicking 3D co-culture system between denuded germinal vesicles with granulosa cells in a barium alginate membrane [83]. They found that the maturation yield of co-cultured oocytes was significantly higher than that in microdrop culture. Combelles et al. reported that co-culture immature oocytes with CCs using 3D collagen gel showed comparable oocytes maturation rates with those cultured only in microdrop [84]. Moreover, pre-culturing the human cumulus-enclosed oocytes [85] or denuded oocytes cocultured with CCs [86] in 3D collagen gel (containing phosphodiesterase 3-inhibitor (PDE3-I) prior to IVM culture showed an improved efficiency of IVM. Collectively, the 3D culture system simulates physiological conditions in vivo and has opened new perspectives on the study of human oocyte IVM. With further development of the 3D culture system, it can be considered a promising method for improving the developmental competence of human oocytes.

## 4. Two Kinds of Culturing Methods

Currently, there is no uniform culturing method for human oocyte IVM [38,94]. In the conventional IVM, immature oocytes were directly cultured to MII oocytes after oocyte retrieval (Figure 1A). However, conventional IVM focuses on nuclear maturation, ignoring the synchronous maturation of cytoplasm [95]. The cytoplasmic maturation might lag behind nuclear maturation, which may not be easy to recognize. However, for a successful IVM, both nuclear and cytoplasmic maturations are required [96]. To better mimic in vivo meiotic arrest, the biphasic IVM culturing system (also known as capacitation IVM (CAPA-IVM)), which includes a pre-IVM culturing period (around 24 h) before the IVM culturing (Figure 1B), has been well developed in animal IVM [97]. The pre-IVM culturing was mainly applied to inhibit resumption of meiosis and promote synchronization between the nuclear and cytoplasmic maturation. During this period, the culturing medium is usually supplemented with cyclic adenosine monophosphate (cAMP) modulators [98,99], cyclic guanosine monophosphate (cGMP) modulators, 3-isobutyl-1-methyl-xanthine (IBMX) [100], or c-type natriuretic peptide (CNP) [101,102]. Among these, CNP-mediated biphasic IVM culturing systems have been shown to improve the IVM of oocytes in animals (mouse, bovine, and goat) [101,102,103,104,105]. CNP-mediated biphasic IVM is also applied in PCOS patients [89,106,107], and patients with gynecologic malignancies [108]. Compared with conventional IVM, it showed an improved IVM rate [107,108] and clinical pregnancy rate [107], and a reduced oocytes degeneration rate [108]. The introduction of this biphasic IVM in clinical practice [89,106,107,108] has been regarded as the most prominent development over recent years [18].

In follicles, CNP is produced in mural granulosa cells and released into the extracellular space and follicular fluid [109,110]. It is an agonist of the natriuretic peptide receptor 2 (NPR2), which is expressed by CCs and mural granulosa cells and is essential for producing cGMP and maintaining meiotic arrest of oocyte [110,111]. However, in preovulatory follicles, the activated amphiregulin/epidermal growth factor receptor (EGFR) signaling pathway could suppress CNP/NPR2/cGMP signaling, resulting in meiotic resumption [112]. Application of CNP to COCs from a wide range of mammal species [101,109,113,114,115] could inhibit meiotic resumption. However, notably, CNP does not inhibit meiotic resumption in isolated oocytes [109,113,114] because oocytes do not express NPR2 [109].

## 5. Time of Oocyte Retrieval

The presence and size of a dominant follicle (DF) are currently utilized as the principal measure for timing oocyte retrieval. In females, the follicular diameter is closely associated with the resumption of oocyte meiosis [116,117]. For a normal menstrual cycle, there is generally only one ovulation when the DF reaches around 18–25 mm in diameter. Other sibling follicles undergo atresia. Whereas, for IVM cycles, most centers prefer to schedule oocyte retrieval when the diameter of DF reaches 10–12 mm or when the endometrial thickness reaches at least 5–6 mm for the PCOS patients without leading follicles [21,118,119,120]. DF size exceeding the upper limit can cause detrimental effects on the sibling immature oocytes, resulting in a decreased quality of the retrieved oocytes [21]. However, in clinical practice, there is no consensus on the best timing (in relation to DF size) for oocyte retrieval in IVM cycles. Both the DF ≤ 14 mm [121] and DF ≤ 10 mm [122] have been proposed as the recommended timing for oocyte retrieval. This could be due to the differences in study participants’ type and priming regimens utilized. Prospective RCTs with more detailed subgroups of DF diameter are warranted to classify the most optimal time of oocyte retrieval.

## 6. Time of In Vitro Culture

Considering different culturing systems, priming regimens, and individuals’ endocrine status, the time of in vitro culture was variable in clinical practice. The IVM culture time of human oocytes is usually set around 24–30 h, which has been shown to achieve IVM rates of 40–60% [59,123]. An increasing culture period may increase the oocyte maturation rate, but it could also lead to poor embryo quality and embryo utilization rate, while a short time culture may result in cytoplasmic immaturity because the cytoplasmic maturation usually lags behind nuclear maturation and is less easily identified [124]. It has been reported that the prolonged in vitro culture time from 24 to 30 h could improve the maturation of rabbit oocyte, especially the ooplasmic maturation [125]. However, the prolonged culture time from 24 to 48 h resulted in aging-like changes in bovine oocytes and significantly lower cleavage rates and blastocyst yields [126]. A 30 h IVM incubation period [127,128] was found to give the best balance between IVM rate and embryological outcome.

## 7. Woman’s Age

Oocyte quality and fertility decline as a woman gets older [129]. Oocytes from aging women exhibited DNA damage [130], meiotic spindle abnormalities, chromosomal misalignment, mitochondrial dysfunction, increased aneuploidy rate [131], and decreased zona pellucida birefringence and thickness [132,133]. A cross-sectional study based on 58 women in an ICSI program also reported that the maturation rates of oocytes were significantly reduced in women over 30 years [133]. Similarly, a study based on 133 patients (aged 1–35 years) who underwent FP by ovarian tissue cryopreservation found that patients between menarche to 25 years could achieve the highest IVM rates, while women ≥30 years and pre-menarche girls (<6 years) obtained extremely low IVM rates (<10%) [134]. The changed oocyte maturation capacity in different ages may stem from the epigenetic changes [135] and differential gene expressions [136,137] of oocytes as they age. Besides, it has been proven that ovaries before puberty contain abnormal follicles with decreased follicle development [138]. Thus, the prepubertal oocytes usually present truncated maturation rates.

## 8. Cryopreservation and Other Physical Factors

Though successful birth after cryopreservation of germinal vesicle (GV)-stage oocytes with IVM has been reported in 1998 [139], concerns regarding higher rates of chromosomal abnormalities of immature oocytes after cryopreservation exist [140]. It has been found that vitrification of immature oocytes before IVM can adversely influence oocyte maturation, viability, and subcellular oocyte quality [87]. A meta-analysis based on 14 studies that assessed maturation rate after vitrification of metaphase I (MI)/ GV oocytes concluded that oocyte vitrification decreased the maturation rate by 24% [141]. The vitrification-warming procedure may play a vital role in the decreased developmental potential of oocytes. Meanwhile, it has been shown that vitrification may reduce oocyte competence by increasing dead oocytes and ROS levels [142]. When the ROS level exceeds the physiological limit, the oocyte developmental competence is compromised [81,143], while the rate of apoptosis is increased [144,145].

On the other hand, it has been reported that aspiration under extremely high pressure may strip the CCs surrounding the oocyte, causing adverse effects on oocyte maturation [146]. Currently, the aspiration pressures range between –80 to –120 mm Hg, and needle sizes range between 16 to 20 gauge [26,147]. However, the optimal pressure and needle size have not yet been clearly established. Other physical factors, such as low oxygen tension [148], short-term temperature elevation [149], and micro-vibration [150], were also investigated in animal experiments. Low oxygen tension (5% O_2_) was found to be beneficial for IVM outcomes [148]. In a previous study, the researchers confirmed that the low oxygen tension contributed to increased levels of tri-methylated-histone-H3-lysine-4 (H3K4me3), indicating active gene transcription [151]. Moreover, short-term temperature elevation (heat stress) can disrupt the developmental competence of the oocytes [149]. Compared with the static system, the micro-vibration culturing system can improve the embryo quality and clinical outcomes of GV-stage oocytes from PCO patients [150].

## 9. Conclusions

The influencing factors of human oocyte IVM are summarized in Figure 2; collectively, the application of a biphasic IVM culturing system, culture medium, and the supplementation have an important influence on IVM outcomes. However, the impact of some factors on IVM remains to be determined, such as different culture medium, different protein sources and hormones in medium, and different follicular priming methods. Their respective roles need to be clarified in more well-designed studies (especially prospective RCTs) in the future.

Since 2021, the ASRM has proposed that IVM is no longer considered experimental [21]. Nonetheless, before IVM can be applied in clinical practice, specific training for clinicians and embryologists is still needed. In the long run, IVM can be considered a promising technology, especially as a relevant addition to oncofertility, offering a safer and more physiological alternative to hormone stimulation.

## Figures and Tables

**Figure 1 biomedicines-09-01904-f001:**
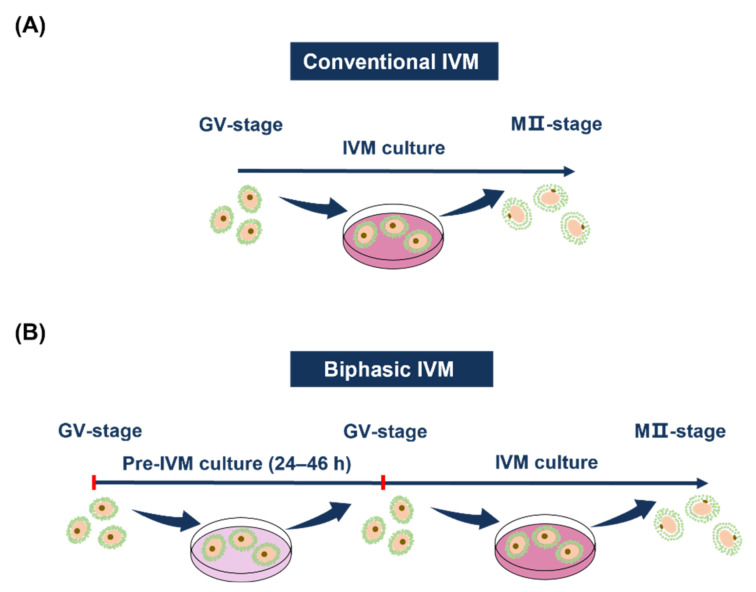
Conventional and biphasic IVM culture systems. (**A**) Conventional IVM system, only includes IVM culture phase. (**B**) Biphasic IVM system, includes a pre-IVM culture phase before IVM culture. The pre-IVM culture inhibits resumption of meiosis in immature oocytes and provides time for acquiring developmental potential. MII, metaphase II.

**Figure 2 biomedicines-09-01904-f002:**
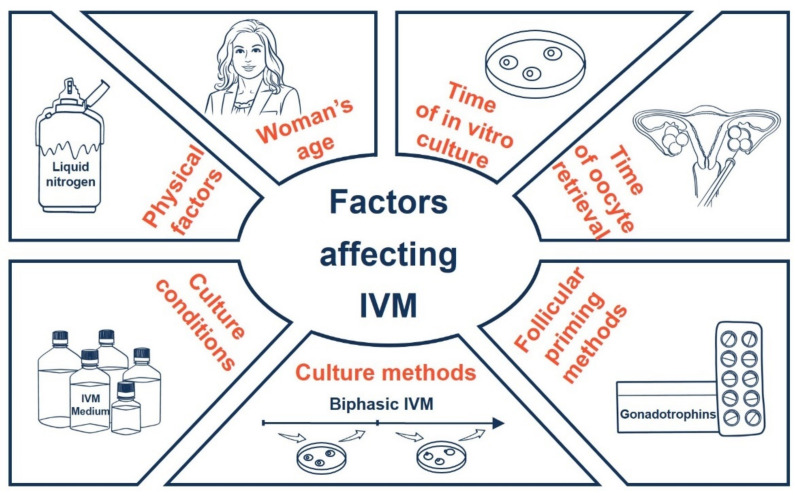
Factors influencing the IVM of human oocytes.

**Table 1 biomedicines-09-01904-t001:** Influence of follicular priming methods on human oocyte in vitro maturation (IVM).

Study	Follicular Priming Methods	Maturation Rate	Fertilization Rate	Clinical Pregnancy Rate	Participants (Samle Size)	Collected Oocytes (Sampe Size)
Mikkelsen et al., 1999 [29]	No stimulation	76%	62%	3/10	Normal cycling women (n = 10)	Oocytes for IVM (n = 37)
	Three days FSH stimulation	85%	65%	2/10	Normal cycling women (n = 10)	Oocytes for IVM (n = 40)
Fadini et al., 2009 [27]	No stimulation	48.4%	77.6%	15.3%	Normo-ovulatory women (n = 100)	Immature COCs (n = 477)
	hCG	57.9%	71.5%	7.6%	Normo-ovulatory women (n = 100)	Immature COCs (n = 442)
						In vivo matured MII (n = 28)
	FSH	50.8%	73.0%	17.3%	Normo-ovulatory women (n = 100)	Immature COCs (n = 461)
	hCG+FSH	77.4%↑	73.0%	29.9%↑	Normo-ovulatory women (n = 100)	Immature COCs (n = 416)
						In vivo matured MII (n = 109)
Mikkelsen et al., 2001 [32]	No stimulation	44%	69%	0	Women with PCOS (n = 9)	Immature COCs (n = 81)
	Three days FSH stimulation	59%↑	70%	29%↑	Women with PCOS (n = 20)	Immature COCs (n = 156)
Wynn et al., 1998 [26]	No stimulation	43.5%	N/A	N/A	Women with healthy ovaries (n = 9)	Immature COCs (n = 46)
	Three days FSH stimulation	71.1%↑	N/A	N/A	Women with healthy ovaries (n = 17)	Immature COCs (n = 114)
Lin et al., 2003 [31]	10,000 IU hCG injection	71.9%	69.5%	36.4%	Women with PCOS (n = 30)	Immature COCs (n = 762)
	Six days FSH stimulation+ 10,000 IU hCG injection	76.5%	75.8%	31.4%	Women with PCOS (n = 30)	Immature COCs (n = 766)
Chian et al., 2000 [30]	No stimulation	69.1%	83.9%	27.3%	Women with PCOS (n = 17)	Immature COCs (n = 81)
	10,000 IU hCG injection	84.3%↑	90.7%	38.5%		Immature COCs (n = 102)

The arrows in the study of Fadini et al. refer to oocyte maturation rate and clinical pregnancy rate in the hCG+FSH group were significantly higher than that in the no stimulation group, hCG group, and FSH group. The arrows in other studies refer to comparisons between groups with or without the corresponding stimulation that were statistically significant. FSH, follicle-stimulating hormone; hCG, human chorionic gonadotrophin; COCs, cumulus-oocyte complexes; MII, metaphase II; PCOS, polycystic ovarian syndrome; N/A, not available.

**Table 2 biomedicines-09-01904-t002:** Influence of culture media on human oocyte IVM.

Study	Medium	Maturation Rate	Fertilization Rate	Clinical Pregnancy Rate	Participants	Age (Years)	Total Sample Size
Cekleniak et al., 2001 [9]	Glucose-free medium (P1)	24 h: 59.7%	N/A	N/A	108 women in ICSI	Mean age: 35.2	369 GV/MI oocytes
		48 h: 71.7%					
	TCM-199	24 h: 44.9%	N/A	N/A			
		48 h: 61.0%					
de Araujo et al., 2009 [39]	TCM-199	82.0%↑	70.0%↑	N/A	13 infertile women with a previous diagnosis of PCOS	26–36	119 immature oocytes
	HTF medium	56.9%	39.4%	N/A			
Pongsuthirak et al., 2014 [48]	IVM-MediCult	65.0%	69.9%	N/A	93 pregnancy women during cesarean deliveries	18–42	1015 immature oocytes
	IVM-SAGE	64.2%	65.2%	N/A			
Filali et al., 2008 [49]	TCM-199	61.0%	61.5%	25%	93 PCOS patients	Mean age: 32.1	1585 immature oocytes
	IVM-MediCult	60.6%	56.8%	28.6%			
Pongsuthirak et al., 2014 [50]	IVM-SAGE	65.0%	66.9%	N/A	89 pregnant women during cesarean deliveries	18-40	1032 immature oocytes
	Blastocyst medium	68.7%	66.4%	N/A			
Fesahat et al., 2017 [51]	Homemade IVM medium	73.3%	54.5%	N/A	220 infertile women	23–37	114 MI oocytes
	Cleavage medium	55.8%	52.6%	N/A			
	Blastocyst medium	72.2%	65.3%	N/A			
	IVM-SAGE	65.5%	63.1%	N/A			
Fesahat et al., 2017 [52]	Homemade IVM medium	55.0%	52.7%	N/A	320 infertile women	Mean age: 31	400 GV oocytes
	Cleavage medium	53.0%	56.6%	N/A			
	Blastocyst medium	78.0%↑	69.0%	N/A			
	IVM-SAGE	68.0%	54.7%	N/A			
Moschini et al., 2011 [53]	Standard culture medium (cleavage medium)	GV→MII: 50.5%	N/A	N/A	28 women in IVF	N/A	127 GV/MI oocytes
		MI→MII: 80.6%					
	IVM-MediCult	GV→MII: 41.2%	N/A	N/A			
		MI→MII: 66.7%					

The arrows in the study of de Araujo et al. refer to the TCM-199 which brought about a significantly higher oocyte maturation rate and fertilization rate than the HTF medium (*p* < 0.01); the arrow in the study of Fesahat et al. refers to the blastocyst medium which brought about a significantly higher oocyte maturation rate than homemade IVM medium (*p* = 0.001) and cleavage medium (*p* < 0.001). TCM-199, tissue culture medium-199; HTF, human tubal fluid; GV, germinal vesicle; MI, metaphase I; IVF, in vitro fertilization.

**Table 3 biomedicines-09-01904-t003:** Influence of medium supplements on human oocyte IVM.

Study	Supplements	Sort	Maturation Rate	Fertilization Rate	Clinical Pregnancy Rate	Culture Medium	Participants	Age (Years)	Total Sample Size
Ashourzadeh et al., 2021 [66]	CCs	Others	ns	↑	N/A	IVM-SAGE	270 women in ICSI	≤35	328 denuded GV oocytes
	GDF9	Cytokine	ns	ns	N/A				
	CCs + GDF9	Cytokine	ns	ns	N/A				
Mohsenzadeh et al., 2019 [87]	GDF9	Cytokine	ns	N/A	N/A	IVM-MediCult	women with cervix and uterine malignancy	21–39	59 denuded frozen-thawed GV/MI oocytes
Chatroudi et al., 2019 [56]	CCs	Others	ns	↑	N/A	IVM medium	women in ICSI	N/A	80 denuded GV oocytes
	GDF9	Cytokine	ns	ns	N/A				
	CCs + GDF9	Cytokine	ns	ns	N/A				
Zhao et al., 2011 [60]	BDNF	Cytokine	↑	N/A	N/A	HTF medium	167 women in ICSI	31 ± 0.3	366 denuded GV/MI oocytes
	GDNF	Cytokine	↑	N/A	N/A				
Cui et al., 2018 [88]	GDNF	Cytokine	↑	ns	N/A	G-IVF medium	82 women in IVF	N/A	200 GV COCs
Ben-Ami et al., 2011 [57]	AREG + EREG	Cytokine	↑	ns	N/A	N/A	30 women in ICSI	20–40	105 GV oocytes with partly remaining cumulus oophorus
Yu et al., 2012 [65]	EGF + BDNF + IGF-1	Cytokine	↑	ns	N/A	IVM medium	women in ICSI	N/A	GV oocytes
	EGF + BDNF + IGF-1	Cytokine	ns	ns	N/A	IVM medium	women in ICSI	N/A	MI oocytes
Goud et al., 1998 [42]	EGF	Cytokine	↑	ns	N/A	Medium 199	38 women in ICSI	mean age 31.9	112 cumulus-denuded GV oocytes
	EGF	Cytokine	ns	↑	N/A	Medium 199	54 women in ICSI	mean age 31.8	177 cumulus-intact GV oocytes
Sánchez et al., 2017 [89]	PMC with CNP+IVM with FSH and AREG	Cytokine	↑	ns	N/A	IVM-MediCult	15 PCOS patients in IVM	N/A	381 immature COCs
Ma et al., 2020 [80]	Coenzyme Q10	Antioxidant	↑	N/A	N/A	IVM medium	45 women in IVF	38-46	92 GV oocytes enclosed by CCs
	Coenzyme Q10	Antioxidant	ns	N/A	N/A	IVM medium	18 women in IVF	≤ 30	74 GV oocytes enclosed by CCs
Li et al., 2019 [41]	Melatonin	Antioxidant	↑	↑	N/A	Medium 199	women in ICSI	N/A	197 denuded GV oocytes
Cao et al., 2020 [45]	Quercetin	Antioxidant	↑	ns	N/A	Medium 199	57 women in IVF	22–42	105 denuded GV oocytes
			ns	ns	N/A				37 denuded GVBD oocytes
Liu et al., 2018 [46]	Resveratrol	Antioxidant	↑	N/A	N/A	Medium 199	64 women in ICSI	38–45	75 denuded GV oocytes
Xie et al., 2021 [90]	Lysophosphatidic acid	Others	↑	N/A	N/A	N/A	43 healthy women with cesarean sections	18–35	155 denuded GV/MI oocytes
Madkour et al., 2018 [91]	Autologous FF	Others	ns	ns	N/A	S-IVM medium (ATL)	47 PCOS patients in IVM	< 40	159 denuded GV/MI oocytes
	Heterologous FF	Others	ns	ns	N/A				
	Heterologous FF + heterologous CGC supernatant	Others	↑	ns	N/A				
Li et al., 2019 [40]	GH	Others	↑	ns	N/A	Medium 199	women in ICSI	N/A	252 denuded GV oocytes
Virant-Klun et al., 2018 [67]	CCs from autologous mature oocytes	Others	↑	N/A	N/A	IVM-MediCult	women in IVF	N/A	174 denuded GV oocytes
Wang et al., 2018 [47]	High-glucose	Others	↓	N/A	N/A	TCM-199	68 women in ICSI	< 35	109 denuded MI oocytes
Anderson et al., 2010 [43]	Blocking antibodies to BDNF	Others	↑	N/A	N/A	T-199 medium	67 women requesting laparoscopic sterilization	21.2–42.9	Immature COCs
Zeng et al., 2013 [92]	Pre-IVM with heparin + cAMP modulators	Others	↓	N/A	N/A	pre-IVM media	42 PCO/PCOS patients in IVM	N/A	Immature COCs
Machtinger et al., 2013 [93]	BPA	Others	↓	N/A	N/A	IVM-SAGE	121 women in IVF/ICSI	23.9–43.9	352 GV oocytes enclosed by some residual cumulus/corona radiata cells

The arrows refer to comparisons between groups with or without the corresponding supplement that were statistically significant. CCs, cumulus cells; GDF9, growth differentiation factor 9; BDNF, brain-derived neurotrophic factor; GDNF, glial cell line-derived neurotrophic factor; AREG, amphiregulin; EREG, epiregulin; EGF, epidermal growth factor; BDNF, brain-derived neurotrophic factor; IGF-1, insulin-like growth factor-1; PMC, prematuration culture; CNP, C-Type Natriuretic Peptide; GVBD, germinal vesicle breakdown; FF, follicular fluid; CGC, cumulus-granulosa cell; GH, growth hormone; cAMP, cyclic adenosine mono-phosphate; BPA, bisphenol-A; ICSI, intracytoplasmic sperm injection; PCO, polycystic ovaries; ns, non-significance.

## Data Availability

Not applicable.

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
