# Peer review of "Factors Influencing the In Vitro Maturation (IVM) of Human Oocyte"

_biomedicines, 2021, doi:10.3390/biomedicines9121904_

Round 1

Reviewer 1 Report

Dear Authors, it is hard to make sense of human IVM as nobody uses the same definition, the same stimulation protocol, same media... It is then even harder to be able to compare IVM between studies. In that sense, I think the title, the paper the structure of the review and the message is a little bit misleading. Unfortunately, you cannot compare culture media as demographic, stimulation protocol, add-ons in media... are so different that it become impossible to identify what is doing what. I think the structure of the paper should be changed starting by the different stimulation protocol and what kind of eggs you get from it. It is what is going to determine what kind of eggs and what kind of culture system need to be used. Bellow you can find some specific comments.

  • Abstract: “In vitro maturation (IVM) of oocytes is a novel and promising assisted reproductive technology (ART) deemed as a simple and safe procedure” IVM is not a novel ART.
  • “it has not been widely used in clinical” It is still not widely used in clinical because it is underperforming
  • Introduction:

“With the development of IVM in recent years, the clinical indications for IVM treatment expanded to (1) fertility preservation (FP) for women at risk of fertility loss and women seeking planned oocyte cryopreservation, (2) high/poor response to exogenous gonadotropin treatment, (3) oocyte donation, (4) ovulatory women with few antral follicle count, (5) preimplantation genetic diagnosis, (6) patients with thrombophilia, and (7) previous failed IVF at-tempts/recurrent implantation failure [7–11].” This would need more explanation, IVM is suboptimal compared to classic stim so why would a doctor request IVM for oocytes donation? Few antral follicles?  PGD (What is the link with IVM?)?... Unclear and misleading please change.

  • “Aside from the inadequate experience of embryologists and gynecologists with this new technology” Again IVM is no new technology it is actually used since the beginning of Human IVF.
  • 1. Culture medium

Were all the culture media mentioned used the same way? Same patient demographic? Same stimulation protocol? Any add-ons in the media? FSH? LH? EGF?... If not, can you really compare the efficacy of these media? Please change

  • 3. Hormones in culture medium

“It is important to note that in most studies, FSH was usually used in combination with LH [23,44–46]/hCG [20,21,24–26,47], which promotes and regulates the maturation of oocyte [48] subsequently improving the IVM outcomes of human oocyte [49].” Please change, it is misleading. LH promotes and regulates maturation through the follicle not the cumulus oocyte complex. Pretty old references. Maybe look for some paper from Jaffe LA.

  • Please discuss the signaling pathway leading to maturation. Why CNP is used in a biphasic culture? EGF? FSH?... why, what are they doing in culture system?

Author Response

Reviewer 1

  • Dear Authors, it is hard to make sense of human IVM as nobody uses the same definition, the same stimulation protocol, same media... It is then even harder to be able to compare IVM between studies. In that sense, I think the title, the paper the structure of the review and the message is a little bit misleading. Unfortunately, you cannot compare culture media as demographic, stimulation protocol, add-ons in media... are so different that it become impossible to identify what is doing what. I think the structure of the paper should be changed starting by the different stimulation protocol and what kind of eggs you get from it. It is what is going to determine what kind of eggs and what kind of culture system need to be used.

Answer: Thank you for your kind comments. To provide a more more objective review, we organized two more tables (please see Table 1-2), and evaluated these studies more carefully and objectively. We have rewrited the Title, Introduction, Medium, and Conclusion section (The revision is in red font).

  • Bellow you can find some specific comments. Abstract: “In vitro maturation (IVM) of oocytes is a novel and promising assisted reproductive technology (ART) deemed as a simple and safe procedure” IVM is not a novel ART. “it has not been widely used in clinical” It is still not widely used in clinical because it is underperforming

Answer: This has been modified in the text (please see lines 14, 16, 17, in red font).

  • Introduction: “With the development of IVM in recent years, the clinical indications for IVM treatment expanded to (1) fertility preservation (FP) for women at risk of fertility loss and women seeking planned oocyte cryopreservation, (2) high/poor response to exogenous gonadotropin treatment, (3) oocyte donation, (4) ovulatory women with few antral follicle count, (5) preimplantation genetic diagnosis, (6) patients with thrombophilia, and (7) previous failed IVF at-tempts/recurrent implantation failure [7–11].” This would need more explanation, IVM is suboptimal compared to classic stim so why would a doctor request IVM for oocytes donation? Few antral follicles?  PGD (What is the link with IVM?)?... Unclear and misleading please change.

Answer: This has been modified in the text (please see lines 38-47, in red font).

  • “Aside from the inadequate experience of embryologists and gynecologists with this new technology” Again IVM is no new technology it is actually used since the beginning of Human IVF.

Answer: This sentence has been deleted. We have rewrited the Introduction section, please see lines 24-30.

  • Culture medium Were all the culture media mentioned used the same way? Same patient demographic? Same stimulation protocol? Any add-ons in the media? FSH? LH? EGF?... If not, can you really compare the efficacy of these media? Please change

Answer: Thank you very much for your comments, we fully agree with you. Indeed, it’s not scientific to compare the findings from same medium among different studies. But for the arrows in Table 1-3, they were statistically significant based on the comparison in each individual study after controlling potential confounders (i.e., their comparisons were carried out in the same study population with the same study protocol). The production of these three tables is not intend to compare different studies, but just for listing and summarizing the published studies on the corresponding topics.

  • Hormones in culture medium “It is important to note that in most studies, FSH was usually used in combination with LH [23,44–46]/hCG [20,21,24–26,47], which promotes and regulates the maturation of oocyte [48] subsequently improving the IVM outcomes of human oocyte [49].” Please change, it is misleading. LH promotes and regulates maturation through the follicle not the cumulus oocyte complex. Pretty old references. Maybe look for some paper from Jaffe LA.

Answer: Thanks to point out this mistake. We have read the papers from Prof. Jaffe carefully, and modified the manuscript accordingly (please see lines 189 and 190, in red font).

  • Please discuss the signaling pathway leading to maturation. Why CNP is used in a biphasic culture? EGF? FSH?... why, what are they doing in culture system?

Answer: This has been modified in the text (please see lines 275-283, in red font).

Thank you again for your valuable insight towards improving the manuscript!

Reviewer 2 Report

The review manuscript entitled "Factors influencing the in vitro maturation (IVM) of human oocyte" is an important overview of the IVM procedure and an interesting dissection of the factors involved. It is a comprehensive review that I welcome and support publication in Biomedicines.

General remarks the authors need to consider:

Argumentation and report to citations does not seem right and should be reanalysed and rewritten. After updated statements regarding recent findings, the authors introduce contrary arguments, that the reader interprets as even more recent. But, with a more attentive look, you realize that those references may be either referring to non-comparable procedures, or having been performed much earlier, and therefore should not validate those contrary arguments. Specifically, please check:

- In 2. Culture conditions 2.2. Protein sources in culture medium: Arguments used from reference 36 (2001 paper) are several times used to counter argue arguments from reference 35 (2008 paper). The reader feels misguided.

- In first paragraph of 2. Culture conditions 2.3. Hormones in culture medium: Arguments starts with statements from references 37 and 38 (2008 and 2011 papers) and end with reference 49 (2000 paper). Again, the reader feels misguided.

Additional more specific remarks the authors need to consider:

- First line of the 1. Introduction: Contrary to what is clearly stated by the authors, IVM should not refer to retrieval of metaphase I oocytes, since in this case oocyte maturation already started in vivo. Such position by the authors is not neutral and does not seem in line with recent consensus.

- In third paragraph of 2. Culture conditions 2.1. Culture medium: It is not relevant to discuss "rescue IVM" here and throughout the manuscript. It should be identified and a non-adequate use of the IVM concept that continuously brings confusion to the IVM field.

- In first paragraph of 2. Culture conditions 2.4. Other supplementation in culture medium: Again, it is not relevant to discuss "rescue IVM" here and throughout the manuscript. The authors even state "rescue-IVM" has been excluded from the definition of IVM. This should be better clarified and dismissed from the beginning of the manuscript. As a non-adequate use of the IVM concept, discussion over technicalities and results with "rescue-IVM" can only bring confusion to an intended updated review on IVM.

In 4. Three kinds of follicular priming methods: Authors state that scholars excluded cycles that used hCG from the IVM definition. Then the section details 4.1. FSH, 4.2 hCG and 4.3 FSH+hCG. Please correct this contradiction. We agree "rescue-IVM" cycles are today excluded from the IVM definition, but not cycles intervened with hCG.

As a conclusion, I suggest the authors to correct these statements and argumentation, rewriting some parts and reference publications in a more logical format, before this review manuscript can be considered for publication in Biomedicines.  

Author Response

Reviewer 2

The review manuscript entitled "Factors influencing the in vitro maturation (IVM) of human oocyte" is an important overview of the IVM procedure and an interesting dissection of the factors involved. It is a comprehensive review that I welcome and support publication in Biomedicines.

General remarks the authors need to consider:

Argumentation and report to citations does not seem right and should be reanalysed and rewritten. After updated statements regarding recent findings, the authors introduce contrary arguments, that the reader interprets as even more recent. But, with a more attentive look, you realize that those references may be either referring to non-comparable procedures, or having been performed much earlier, and therefore should not validate those contrary arguments. Specifically, please check:

  • In 2. Culture conditions 2.2. Protein sources in culture medium: Arguments used from reference 36 (2001 paper) are several times used to counter argue arguments from reference 35 (2008 paper). The reader feels misguided.

Answer: Thank you so much for the valuable comments! This has been modified in the text (please see lines 145-167, in red font).

  • In first paragraph of 2. Culture conditions 2.3. Hormones in culture medium: Arguments starts with statements from references 37 and 38 (2008 and 2011 papers) and end with reference 49 (2000 paper). Again, the reader feels misguided.

Answer: The paragraph discussing reference 49 has been deleted.

  • Additional more specific remarks the authors need to consider: - First line of the 1. Introduction: Contrary to what is clearly stated by the authors, IVM should not refer to retrieval of metaphase I oocytes, since in this case oocyte maturation already started in vivo. Such position by the authors is not neutral and does not seem in line with recent consensus.

Answer: The IVM definition has been modified (please see lines 52-57, in red font).

  • In third paragraph of 2. Culture conditions 2.1. Culture medium: It is not relevant to discuss "rescue IVM" here and throughout the manuscript. It should be identified and a non-adequate use of the IVM concept that continuously brings confusion to the IVM field.

 Answer: Sure, we fully agree with you! We have deleted the content of "rescue-IVM" throughout the manuscript.

  • In first paragraph of 2. Culture conditions 2.4. Other supplementation in culture medium: Again, it is not relevant to discuss "rescue IVM" here and throughout the manuscript. The authors even state "rescue-IVM" has been excluded from the definition of IVM. This should be better clarified and dismissed from the beginning of the manuscript. As a non-adequate use of the IVM concept, discussion over technicalities and results with "rescue-IVM" can only bring confusion to an intended updated review on IVM.

 Answer: We have deleted the content of "rescue-IVM" throughout the manuscript.

  • In 4. Three kinds of follicular priming methods: Authors state that scholars excluded cycles that used hCG from the IVM definition. Then the section details 4.1. FSH, 4.2 hCG and 4.3 FSH+hCG. Please correct this contradiction. We agree "rescue-IVM" cycles are today excluded from the IVM definition, but not cycles intervened with hCG.

Answer: Thank you very much for your kind comments! Sure, to date, the definition of IVM is still rather controversial. But we feel that, we are writing a review, so we should discuss the opinions from different scholars… Based on the most recent document from American Society for Reproductive Medicine [1], they also discussed three follicular priming methods in their paper…we finally decided to include the content of hCG in this review.. In the rescue-IVM, the immature oocytes were collected from conventional hyperstimulated IVF/ICSI cycles with HCG triggering…However, in IVM, the follicular priming methods are mild..

Nevertheless, the adverse effects of hCG priming have been discussed in details in the revised manuscript (please see lines 85-100).

Reference:

  1. In vitro maturation: a committee opinion. Fertil. Steril. 2021, 115, 298–304, doi:10.1016/j.fertnstert.2020.11.018.

Reviewer 3 Report

I believe that it is worthy of publication on Biomedicines.

Author Response

Reviewer 3

I believe that it is worthy of publication on Biomedicines.

Answer: Thank you very much for your approval!

Reviewer 4 Report

Thank you for the invited review for “Factors influencing the in vitro maturation (IVM) of human oocyte”

The author will design contents for review and summary of in vitro maturation study for ART. 
IVM is very useful for poor ovarian responders et al. But there is still development media for IVM of human immature oocytes. And author presents clinical media and methods. 
Therefore author shows good organization and writing for an acceptable manuscript.

However, I have some suggest make a better manuscript
The author has written culture conditions as medium protein, hormone et al in section 2. 
I suggest section 2 add one more thing by carbon source like glucose, glutamine, pyruvate et al for IVM
And I have one more suggestion that what about the outcome of commercial media in the clinical data. It is good guidance for embryologist. 
I saw table 1 but there is no clinical outcome after IVM.

Author Response

Reviewer 4

Thank you for the invited review for “Factors influencing the in vitro maturation (IVM) of human oocyte”

The author will design contents for review and summary of in vitro maturation study for ART. 
IVM is very useful for poor ovarian responders et al. But there is still development media for IVM of human immature oocytes. And author presents clinical media and methods. 
Therefore author shows good organization and writing for an acceptable manuscript.

However, I have some suggest make a better manuscript

(1)  The author has written culture conditions as medium protein, hormone et al in section 2. I suggest section 2 add one more thing by carbon source like glucose, glutamine, pyruvate et al for IVM
Answer: Thank you so much for this valuable comment! We have added “3.4. Carbohydrates in culture medium” into the manuscript (please see lines 191-211).

(2) And I have one more suggestion that what about the outcome of commercial media in the clinical data. It is good guidance for embryologist. I saw table 1 but there is no clinical outcome after IVM.

Answer: We have added the clinical pregnancy rate into this Table (please see line 241), meanwhile, in the other two new tables, we also list the clinical pregnancy rate. In general, for the study on the influence of the novel supplement on human oocyte IVM outcomes, the supplementation-treated oocytes were not used for embryo implantation, so the clinical pregnancy rate is missing. There is only one study, from Sánchez et al. [1], reported the clinical pregnancy rate of their negative control group (i.e., standard IVM without special supplementation in culture medium..).

Reference: 1.         Sánchez, F.; Lolicato, F.; Romero, S.; De Vos, M.; Van Ranst, H.; Verheyen, G.; Anckaert, E.; Smitz, J.E.J. An improved IVM method for cumulus-oocyte complexes from small follicles in polycystic ovary syndrome patients enhances oocyte competence and embryo yield. Hum. Reprod. 2017, 32, 2056–2068, doi:10.1093/humrep/dex262.

Thank you again for your valuable comments towards improving the manuscript!

Round 2

Reviewer 1 Report

Dear Author, this 2nd version is way better than the 1st one. The modification you made help understand better the challenges surrounding IVM. Best of luck. 

Author Response

Dear Author, this 2nd version is way better than the 1st one. The modification you made help understand better the challenges surrounding IVM. Best of luck. 

Answer: Thank you very much for your effort.

Reviewer 2 Report

The review manuscript now entitled "Progress report: oocytes in vitro maturation (IVM) and it’s influencing factors" in this revised version is, as previously stated, an important overview of the IVM procedure and an interesting dissection of the factors involved. It is a comprehensive review that I welcome and support publication in Biomedicines.

It is now a better version as many of the remarks from this reviewer were considered in this new version. There are however some aspects the authors still need to evaluate:

  1. The Title. I am not aware of the reasons why the authors changed the title, but I have to say I would prefer the initial title "Factors influencing the in vitro maturation (IVM) of human oocyte" over the current one, which is not so clear and weakens the manuscript.
  2. The new sentences in the manuscript, highlighted in red, seem to need more careful english revision. Just as an example in 1. Introduction, 3rd line of the second paragraph (line 34) should read instead: "For the patients, this technology is cost-effective, safe and convenient". 
  3. There are remains of my previous remarks whereas argumentation and report to citations does not seem right and should be reanalysed and rewritten. After updated statements regarding recent findings, the authors introduce contrary arguments, that the reader interprets as even more recent. But, with a more attentive look, you realize that those references may be either referring to non-comparable procedures, or having been performed much earlier, and therefore should not validate those contrary arguments. Specifically, please check 2. Follicular priming methods and the collected eggs (page 2), second paragraph (lines 75-85), as the message that FSH stimulation may not be needed is based on two 1998 and 1998 papers, questioning findings from more recent papers, namely a 2021 position paper.
  4. Some modifications in the revised version may not be highlighted with the red colour, as the ones identified in section 6. Time of in vitro culture: "and is less easily identified" (line 308) and the last sentence, which was shortened, so it should be highlighted (lines 312-314).
  5. The 9. Conclusions section lost important messages, such as the one that IVM may be a relevant addition to oncofertility, and gained irrelevant messages, such as the one on the need to have "patients’ informed consent about the inferior performance of IVM compared with standard IVF". This detail does not seem necessary since in ART, patients should have informed consent regarding the overall techniques, with no need for this level of detail which can vary according to multiple factors. Should we advise to have specific informed consent on the better results regarding the transfer of blastocysts over cleavage stage embryos? Should we advise to have specific informed consent on the better results on blastocyst transfer in a subsequent frozen cycle over a fresh cycle? This level of detail and knowledge, while important for the professional and eventual explanations to the patient, but does not seem essential to present upfront in a patient signed informed consent for example.

Author Response

Reviewer 2

The review manuscript now entitled "Progress report: oocytes in vitro maturation (IVM) and it’s influencing factors" in this revised version is, as previously stated, an important overview of the IVM procedure and an interesting dissection of the factors involved. It is a comprehensive review that I welcome and support publication in Biomedicines.

It is now a better version as many of the remarks from this reviewer were considered in this new version. There are however some aspects the authors still need to evaluate:

  • The Title. I am not aware of the reasons why the authors changed the title, but I have to say I would prefer the initial title "Factors influencing the in vitro maturation (IVM) of human oocyte" over the current one, which is not so clear and weakens the manuscript.

Answer: Thank you very much for the title suggested. We have modified it accordingly (please see line 2).

  • The new sentences in the manuscript, highlighted in red, seem to need more careful english revision. Just as an example in 1. Introduction, 3rd line of the second paragraph (line 34) should read instead: "For the patients, this technology is cost-effective, safe and convenient".

Answer: Thank you very much for your careful review, and sorry for our English grammar error! We have corrected this sentence (please see line 32). Meanwhile, the whole manuscript has been carefully proofread again to avoid typos and grammatical errors. The modifications could be seen throughout the manuscript marked up using the “Track Changes” function in Word.

  • There are remains of my previous remarks whereas argumentation and report to citations does not seem right and should be reanalysed and rewritten. After updated statements regarding recent findings, the authors introduce contrary arguments, that the reader interprets as even more recent. But, with a more attentive look, you realize that those references may be either referring to non-comparable procedures, or having been performed much earlier, and therefore should not validate those contrary arguments. Specifically, please check 2. Follicular priming methods and the collected eggs (page 2), second paragraph (lines 75-85), as the message that FSH stimulation may not be needed is based on two 1998 and 1998 papers, questioning findings from more recent papers, namely a 2021 position paper.

Answer: Thank you for your careful review and thoughtful comments. We have re-analyzed and re-written the “2. Follicular priming methods and collected eggs, second paragraph (please see lines 87-104)”.

  • Some modifications in the revised version may not be highlighted with the red colour, as the ones identified in section 6. Time of in vitro culture: "and is less easily identified" (line 308) and the last sentence, which was shortened, so it should be highlighted (lines 312-314).

Answer: Thank you for the careful review, we have highlighted them using “Track Changes” function in Word (please see line 458 and line 464).

  • The 9. Conclusions section lost important messages, such as the one that IVM may be a relevant addition to oncofertility, and gained irrelevant messages, such as the one on the need to have "patients’ informed consent about the inferior performance of IVM compared with standard IVF". This detail does not seem necessary since in ART, patients should have informed consent regarding the overall techniques, with no need for this level of detail which can vary according to multiple factors. Should we advise to have specific informed consent on the better results regarding the transfer of blastocysts over cleavage stage embryos? Should we advise to have specific informed consent on the better results on blastocyst transfer in a subsequent frozen cycle over a fresh cycle? This level of detail and knowledge, while important for the professional and eventual explanations to the patient, but does not seem essential to present upfront in a patient signed informed consent for example.

Answer: Thank you so much for the insightful comments! We have added the important message associated with oncofertility in lines 521-522, and deleted the messages associated with patients’ informed consent (please see lines 519-520).

Reviewer 4 Report

Great revision for publishing. Most of the comments are well updated in the revised manuscript. 

It is acceptable. 

Author Response

Reviewer 4

Great revision for publishing. Most of the comments are well updated in the revised manuscript. It is acceptable.

Answer: Many thanks for your appreciation of our work.